# Analysis of Long Non-Coding RNA (lncRNA) UCA1, MALAT1, TC0101441, and H19 Expression in Endometriosis

**DOI:** 10.3390/ijms231911583

**Published:** 2022-09-30

**Authors:** Tomasz Szaflik, Hanna Romanowicz, Krzysztof Szyłło, Radosław Kołaciński, Magdalena M. Michalska, Dariusz Samulak, Beata Smolarz

**Affiliations:** 1Department of Gynecology, Oncological Gynecology and Endometriosis Treatment, Polish Mother’s Memorial Hospital Research Institute, Rzgowska 281/289, 93-338 Lodz, Poland; 2Laboratory of Cancer Genetics, Department of Pathology, Polish Mother’s Memorial Hospital Research Institute, Rzgowska 281/289, 93-338 Lodz, Poland; 3Department of Obstetrics and Gynecology and Gynecological Oncology, Regional Hospital in Kalisz, 62-800 Kalisz, Poland; 4Department of Obstetrics, The President Stanisław Wojciechowski Calisia Academyin, 62-800 Kalisz, Poland

**Keywords:** lncRNA, UCA1, MALAT1, TC0101441, H19, endometriosis

## Abstract

Endometriosis is a disease of complex etiology. Hormonal, immunological, and environmental factors are involved in its formation. In recent years, special attention has been paid to genetic mechanisms that can have a significant impact on the increased incidence of endometriosis. The study aimed to analyze the expression of four long non-coding RNA (lncRNA) genes, UCA1, MALAT1, TC0101441, and H19, in the context of the risk of developing endometriosis. The material for genetic testing for the expression of lncRNA genes were tissue slices embedded in paraffin blocks from patients with endometriosis (*n* = 100) and the control group (*n* = 100). Gene expression was determined by the RT-PCR technique. The expression of the H19 gene in endometriosis patients was statistically significantly lower than in the control group. A statistically significant association was found between H19 gene expression in relation to The Revised American Society for Reproductive Medicine classification of endometriosis (rASRM) in the group of patients with endometriosis. Research suggests that H19 expression plays an important role in the pathogenesis of endometriosis.

## 1. Introduction

Endometriosis is an estrogen-dependent, gynecological disease that, due to the accompanying ailments and chronic nature, is a very important medical, social and economic problem [1].

For endometriosis, subtypes of endometriosis, treatment, and outcomes, different definitions are used. This has important implications for research and clinical practice. A paper prepared by the International Working Group of AAGL, ESGE, ESHRE, and WES presented a list of 49 terms used in endometriosis; including the definition of endometriosis and its subtypes, different locations, interventions, symptoms, and outcomes [2]. It will also provide an appropriate definition of endometriosis. Endometriosis is defined as an inflammatory process characterized during surgery by the presence of an epithelium and/or endometrial-like stroma outside the endometrium and myometrium, usually accompanied by an inflammatory process.

Recently, in the context of endometriosis, long non-coding RNAs (lncRNAs) have become of interest, which covers more than 200 bp and are a subtype of non-coding RNAs (ncRNAs) [3,4,5,6,7].

Unlike a group of short, non-coding RNAs (sncRNAs) such as microRNAs (miRNAs), long non-coding RNAs tend to show greater sequence matching and, thus, specificity of action relative to target genes [8].

Non-coding RNA molecules participate in the processes of regulation at virtually all stages of the transmission of genetic information: from DNA to protein. Particularly spectacular is the involvement of certain non-coding RNA molecules in the mechanisms leading to the switching on or off of the expression of individual genes. In the research of Zhou et al., 388 of the lncRNA transcripts tested showed overexpression, and 188 showed reduced levels of expression in the ectopic endometrium compared to the eutopic endometrium [3]. It is known that the expression of many lncRNA is subject to changes in (a) the serum of women with endometriosis compared to healthy women, (b) the eutopic endometrium of women with endometriosis compared to healthy women, (c) in the ectopic endometrium of the ovaries compared to the eutopic endometrium in women with endometriosis [4].

Recent literature data suggest that the following lncRNAs are associated with endometriosis: UCA1, MALAT1, TC0101441, and H19 [5,6]. In the work of Mejer et al., it was presented that, specifically, these four lncRNAs play an important role in pathogenesis and the development of endometriosis [6].

Although the above lncRNA sequences are somehow correlated with endometriosis, the exact meaning and context of this interaction remain unclear and need to be explained. This work is designed to expand the still vague and incomplete knowledge of the role of selected lncRNA in endometriosis. There is limited literature data on these lncRNAs, and until now, none of them has been studied in endometriosis in the Polish population.

In our study, we presented an analysis of the expression of UCA1, MALAT1, TC0101441, and H19lncRNAs in endometriosis patients compared to controls, to possibly correlate these lncRNAs with the risk of endometriosis.

## 2. Results

The patient’s characteristics are shown in Table 1. Patients with endometriosis were between the ages of 23 and 58 (mean age 36.4 ± 6.0). Control patients were aged 29 to 61 years (mean age 38.3 ± 6.2). The clinical stage of patients with endometriosis was determined according to the rASRM (The Revised American Society for Reproductive Medicine) classification of endometriosis 1996) [9].

H19, UCA1, MALAT1, and TC0101441 expression results in endometriosis patients and controls, measured by relative quantification, are presented in Figure 1, Figure 2, Figure 3 and Figure 4, respectively.

There were no statistically significant differences in the expression of UCA1, MALAT1, and TC0101441 between patients and the control group. Statistically, there was a significantly lower expression of H19 in cases of endometriosis compared to controls.

Table 2 presents the comparisons of expressions of the studied lncRNA in patients and controls.

The expression of UCA1, MALAT1, TC0101441, and H19 lncRNA sequences were also statistically analyzed for correlation with clinical-pathological data on age, parity, spontaneous abortions, hormone replacement therapy, and BMI. All the correlations listed above turned out to be statistically insignificant. lncRNA expression was also analyzed statistically for correlation with the clinical stage of endometriosis. An association was observed between H19 gene expression and rASRM classification in the endometriosis group (Table 3).

## 3. Discussion

Despite the fact that research on biomarkers of endometriosis is still ongoing, there are still no satisfactory results, which makes it impossible to carry out effective laboratory diagnostics used in the diagnosis and monitoring of the treatment of the disease [10].

Recently, in the context of endometriosis, long non-coding RNAs have become the subject of interest. However, the clinical significance and biological mechanism of lncRNA in the development of endometriosis remain largely unknown. There are reports in the scientific press suggesting a link between lncRNA expression and the development of endometriosis [5,6,7,11,12,13,14,15].

This study aimed to analyze the level of expression of four long non-coding RNA (UCA1, MALAT1, TC0101441, and H19) in endometriosis.

In the context of lncRNA expression levels, the following were evaluated: clinical stage of endometriosis, age, BMI, parity, and spontaneous abortions. An interesting aspect of such an assessment is whether lncRNA can serve as a biomarker in a group of patients affected by endometriosis. On this basis, it is possible to identify patients belonging to the risk group whose prognosis would be worse and the therapeutic process should be more aggressive—even in the case of endometriosis at an early clinical stage.

The only lncRNA in which our statistical analysis showed a significant correlation with endometriosis was H19. In cases of the disease, a reduced level of H19 expression was shown. In addition, H19 levels correlated with the clinical degree (staging) of endometriosis.

There are reports similar to our results. Ghazal et al. showed that H19 expression was reduced in eutopic endometrial patients with endometriosis compared to the control group [16]. Reduced H19 expression leads to increased let-7 bioavailability, which in turn inhibits IGF1R expression at the post-transcriptional level, thus, contributing to reduced stromal cell proliferation. Disruption of the H19/Let-7/IGF1R regulatory pathway may contribute to impaired endometrial preparation and receptivity in women with endometriosis.

Liu et al. showed that H19 lncRNA was downregulated in mononuclear cells from peritoneal fluid (PFMC) in endometriosis patients [17]. Korucuoglu et al. presented that H19 expression is reduced in endometrial tissues of infertile women for an unexplained cause [18]. There are studies showing that the expression of H19 lncRNA in the ectopic and eutopic endometrial tissues of endometriosis is much higher than in normal endometrial tissues [19,20,21]. H19 overexpression in endometriosis lesions is associated with infertility, relapse, bilateral ovarian lesions, elevated CA125 levels, and progression in the altered stage of the American Fertility Society disease (rAFS). Multivariate logistic regression analysis showed that H19 overexpression in endometriosis lesions is a prognostic factor for endometriosis recurrence [21]. Further studies of H19 expression in endometriosis showed that expression could be differentiated in endometriosis. In the case of endometriosis tissues and/or cells, both up- and downregulated H19 were observed. For body fluids or lesion microenvironment H19 was downregulated [5]. The discrepancy in the researchers’ H19 expression results is likely the result of differences in race and perhaps the stage of the disease. Further genetic testing is, therefore, needed.

Endometriosis is known to be an estrogen-dependent disease. Therefore, lncRNAs involved in the estrogen pathway or those directed by estrogen signaling may play a role in this disease. H19 is positively regulated by estrogen, and its expression in the endometrium increases in the proliferative phase of the menstrual cycle [18]. H19 has been shown to regulate several pathways that are important in endometriosis, IGF1R, ITGB3, IER3, and ACTA2 [16,17,20,22]. H19 can act as a miRNA sponge in endometriosis. Reduced H19 levels have been shown to be associated with an increase in let-7 miRNA activity. The consequence is the inhibition of IGF1R expression, which results in reduced proliferation of endometrial stromal cells [20]. The H19/let7/IGF1R pathway may contribute to impaired endometrial receptivity in women suffering from this disease. H19 regulates cell proliferation and invasion of endometrial ectopic cells by increasing ITGB3 expression through the miR-124-3p sponge [17]. H19 is also involved in the impaired immune response of women with the disease, acting as a sponge for miR-342-3p, which regulates the IER3 pathway. This pathway is involved in the differentiation of Th-17 cells and the proliferation of endometrial stromal cells in ectopic sites in women with this disease.

In our study, three of the four sequences we studied (UCA1, MALAT1, TC0101441) were found to be statistically insignificant from the point of view of endometriosis risk. Our results are in contrast to the literature data that support the role of these lncRNAs in endometriosis.

Long non-coding RNA urothelial carcinoma-associated 1 (UCA1) has been proven to be involved in the pathogenesis of various diseases in humans, including various types of ovarian diseases, such as ovarian cancer [23]. Recent research indicates that UCA1 participates in the pathogenesis of endometriosis [24].

Research by the team of Huang et al. showed that in most patients, the level of expression of lncRNA UCA1 was significantly higher in the ectopic endometrial tissues than in paired eutopic endometrial tissues [24].

Compared with healthy controls, serum lncRNA-UCA1 levels were lowered in patients with ovarian endometriosis, and serum UCA1 levels decreased further as the disease progressed. Serum levels of lncRNA-UCA1 l showed no significant correlation with either the age of the patients or their life habits. After treatment, serum UCA1 levels increased, and serum UCA1 levels on the day of discharge were significantly lower in relapsed patients than in non-relapse patients [24].

Based on these data, it can be concluded that the downregulation of UCA1 is involved in the pathogenesis of ovarian endometriosis and may be a promising diagnostic and prognostic biomarker of this disease. Since endometriosis is known as a precursor to some types of ovarian cancer, similar biological functions of lncRNAs detected in cancer may also determine the development of endometriosis [25].

UCA1 can interact with various miRNAs, which then change gene expression profiles and control tumor progression. The participation of genetic variants of UCA1 in the development of endometriosis and related infertility is indicated [26].

It is known that UCA1 may be involved in the occurrence and development of endometriosis by promoting cell proliferation and inhibiting apoptosis [27].

MALAT1 expression is enhanced in many cancer tissues, and MALAT1 is involved in the proliferation, apoptosis, migration, invasion, and spread of cancer cell metastasis [28]. Endometriosis can be considered a mild metastatic disease and, in addition, due to the ability of endometrial tissue to infiltrate, form metastases and relapses, like tumors, it is very similar to cancer [29]. Epidemiological data suggest that endometriosis has malignant potential [30]. MALAT1 lncRNA expression has been shown to be significantly elevated in endometrial ectopic tissues compared to eutopic endometrial tissues [31].

Yu et al. found that MALAT1 lncRNA can promote endometrial cell apoptosis and regulate MMP-9 expression through the NF-κB/iNOS pathway, thus, mediating the pathogenesis of endometriosis [32].

MALAT1 levels were significantly higher in ectopic endometrial tissues than in paired eutopic endometrial tissues [33].

lncRNA-MALAT1 can also work by regulating autophagy under hypoxia conditions in endometriosis. MALAT1 silencing has been shown to inhibit hypoxia-induced autophagy in human endometrial stromal cells [34].

TC0101441 is a newly identified lncRNA associated with cancer metastasis [35,36].

Since endometriosis has a prometastatic effect similar to those observed in cancer, it was decided to determine the role of TC0101441 in this disease. Analyses TC0101441 concentration in extracellular vesicles contributes to migration/invasion of endometrial cyst stromal cells (ECSC) [37]. It was shown that lncRNA-TC0101441 is strongly expressed in ectopic lesions and that serum extracellular follicular TC0101441 levels were significantly increased in patients with stage III/IV endometriosis compared to patients with stage I/II endometriosis and control. This indicates the potential of extracellular vesicular TC0101441 as a biomarker of endometriosis [38].

lncRNA-TC0101441 in epithelial ovarian cancer (EOC) promotes EOC cell migration/invasion [39].

Extracellular transport of TC0101441 via exosomal vesicles is known to enhance the migration and invasion of endometriosis [39].

All the literature reports cited above encourage the conclusion that the study of selected lncRNA sequences is supported by convincing premises, and research in this field must be continued.

The results presented in this paper on the analysis of lncRNA genes reveal the existence of certain relationships of their expression with endometriosis. It should be noted, however, that the presented research covered a small population (100 patients and 100 control) and requires further work carried out on much larger groups of respondents. The study groups we use may simply be quantitatively unsatisfactory to draw the right conclusions. Second, the cases and controls are not exactly homogeneous—to some extent, they differ in age, spontaneous abortion, and the use of hormone replacement therapy, which can falsify the results. What is more, the cases are not a group of disease-free women, but they were all treated surgically for a mild gynecological condition—symptomatic uterine fibroids. Some reports suggest that some lncRNA sequences (i.e., lncRNA *H19*) may exhibit a correlation with uterine fibroids’ tissue as such [40], yet in our study, the genetic assays were performed strictly on the selected endometrial tissues and not on uterine fibroids.

With all of the above findings in mind and realizing the limitations of our study, we dare say that this research has shed new light on lncRNAs in endometriosis and contributes to a growing—but still unclear—knowledge of these non-coding sequences in endometriosis.

## 4. Materials and Methods

### 4.1. Patients

The study group consisted of 100 patients diagnosed with endometriosis in the laparoscopic and pathological examination. On the other hand, the control group consisted of 100 patients who had no endometriosis during the surgical procedure, and the eutopic endometrium was normal in the histopathological examination. These patients were operated on for uterine fibroids. The exclusion criterion was concomitant cancer in patients pre-classified to the study group and existing cancers in patients from the control group. Patients were selected for studies in the period between 2015 and 2016. The material for analysis was RNA isolated from paraffin blocks obtained from material collected during surgery or curettage of the uterine cavity in the Department of Gynecology, Gynecology Oncology and Endometriosis Treatment of the Polish Mother’s Memorial Hospital Research Institute, Lodz. Paraffin blocks came from the archives of the Department of Clinical Pathology of the Polish Mother’s Memorial Hospital Research Institute, Lodz. All preparations have been characterized histologically. A formal consent (No. 98/2015) was obtained from the Bioethical Committee of the Institute-Polish Mother’s Memorial Hospital in Lodz, Poland.

### 4.2. RNA Isolation

Total RNA was isolated from formalin-fixed paraffin-embedded (FFPE) tissues using the High Pure RNA isolation kit (Roche Diagnostics GmbH, Mannheim, Germany), according to the manufacturer’s instructions. The FFPE samples were placed in 2 mL Eppendorf tubes, dewaxed with 100% xylene, washed in 100% ethanol, and dried at 55 °C for 10 min. The dried tissue was suspended in 100 μL of paraffin tissue lysis buffer (included in the kit) and digested with proteinase K at 55 °C overnight. The resulting total RNA was used for cDNA synthesis or stored at −80 °C until use.

### 4.3. Spectrophotometric Analysis of Purity and Concentration of RNA

The purity of the obtained RNA preparations was determined by the spectrophotometric method by twice measuring the absorbance of each sample at wavelengths of 260 nm and 280 nm. The adopted criterion for RNA purity was A260/A280 within 1.8–2.0. The concentration of RNA was determined by the spectrophotometric method based on the absorbance value measured at a wavelength of 260 nm. This value corresponded to the following relationship:1OD = 40 µg RNA/mL

The mean values of purity and RNA concentration obtained for the test material and the correct one met the necessary criteria.

### 4.4. RT-PCR Reaction

The reverse transcription was carried out using the Maxima First Strand cDNA synthesis kit (ThermoFisher Scientific, Inc., Waltham, MA, USA) according to the manufacturer’s protocols. 500 ng of total RNA was used as the starting material. Reverse transcription was performed under conditions optimized for use with this kit (25 °C for 10 min, 50 °C for 30 min, 85 °C for 5 min). The cDNA samples were stored frozen at −20 °C.

Quantification of lncRNA was carried out using TaqMan™ (Applied Biosystems, Lincoln Centre Dr, Foster City, CA, USA). In addition, GADPH Assay has been used as an endogenous control. qPCR reactions were performed in a volume/reaction of 10 μL, including 10 ngcDNA, 5 μL TaqMan Fast Advanced PCR Master Mix, and 0.5 μL of a suitable primer (20×) (OriGene Technologies GmbH, Schillerstr.532052, Herford, Germany). The expression of GAPDH served as an endogenous control. The following primer sequences were used: TC0101441: forward, 5′ AAGGCAGGTGAGAACGAGT3′; reverse, 5′CTCGACTTAGGGAGCTGCAC3′, UCA1: forward 5′-TTTATGCTTGAGCCTTGA-3′; reverse 5′-CTTGCCTGAAATACTTGC-3′.

MALAT1: forward 5′ GACTTCAGGTCTGTCTGTTCT3′; reverse 5′ CAACAATCACTACTCCAAGC3′, H19: forward 5′ ATCGGTGCCTCAGCGTTCGG3′; reverse 5′ CTGTCCTCGCCGTCACACCG3′ and GAPDH: forward, 5′ AGCCACATCGCTCAGACAC3′; reverse 5′ GCCCAATACGACCAAATCC3′. Each assay was performed in triplicate. The samples were incubated in a 96-well plate at 95 °C for 3 min, followed by 40 cycles of 95 °C for 1 s and 60 °C for 20 s. The relative level of expression was determined by the 2-ΔΔCq method.

### 4.5. Statistical Analysis

Statistical analysis of the obtained results was carried out using the STATISTICA 11 program (StatSoft, Krakow, Poland). The analysis of the significance of differences in the level of gene expression at the mRNA level was carried out using non-parametric tests (Mann–Whitney U test, Kruskal–Wallis test) due to the lack of normality of the distribution of the obtained results, which was confirmed by the Shapiro–Wilk test. The correlation analysis between the variables was performed using the R-Spearman test. Statistical significance was observed at *p* < 0.05.

## 5. Conclusions

In view of the relevance of the results obtained, these studies should be continued. The results obtained in the work contribute to the broadening of knowledge on the subject of molecular mechanisms conducive to the development of endometriosis. The study showed differences in lncRNA expression between the studied groups of patients that could potentially promote the development of the disease. Due to the small number of studies on lncRNA expression analyses in patients with endometriosis in the Polish population, the effect of the research makes a significant contribution to expanding knowledge about the impact of genetic factors on the development of endometriosis. Understanding the relationship between gene expression and endometriosis may contribute to the development of new therapeutic strategies in the context of this disease. Gene expression testing could be a future target for personalized therapy.

The current state of knowledge about lncRNA in endometriosis is still limited. Further research is warranted to further explore this topic.

## Figures and Tables

**Figure 1 ijms-23-11583-f001:**
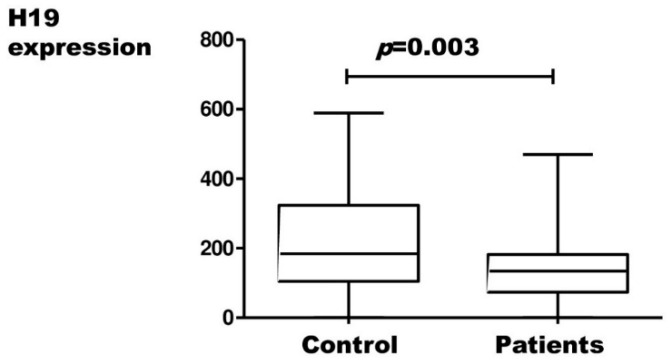
Expression of *H19* in endometriosis patients and controls (Mann–Whitney test; *p* < 0.05).

**Figure 2 ijms-23-11583-f002:**
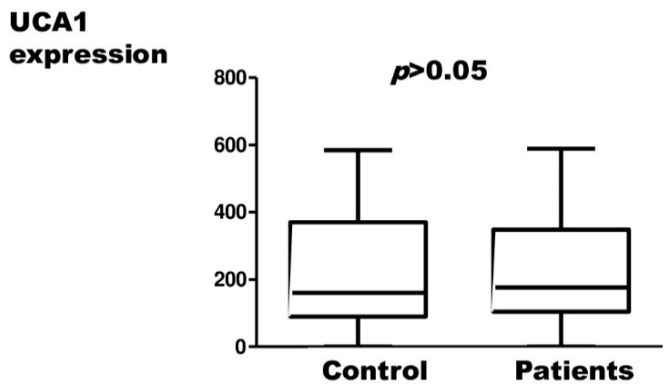
Relative expression of the UCA1 in endometriosis patients with respect to the control material (Mann–Whitney test; *p* > 0.05).

**Figure 3 ijms-23-11583-f003:**
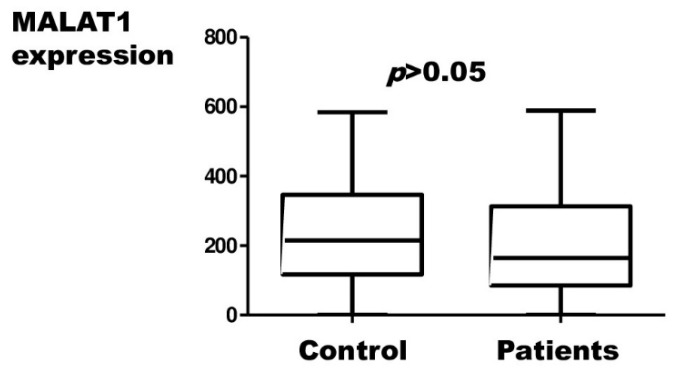
Relative expression of the MALAT1 in endometriosis patients with respect to the control material (Mann–Whitney test; *p* > 0.05).

**Figure 4 ijms-23-11583-f004:**
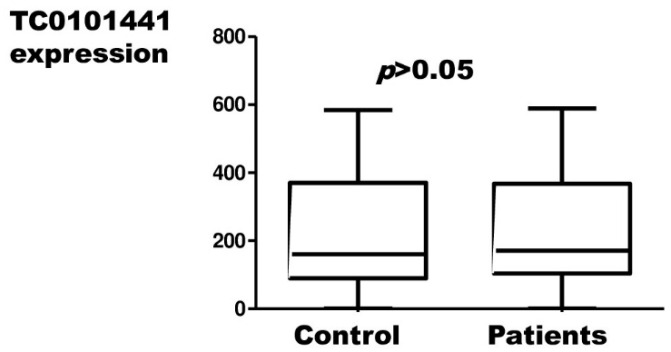
Relative expression of the TC0101441 in endometriosis patients with respect to the control material (Mann–Whitney test; *p* > 0.05).

**Table 1 ijms-23-11583-t001:** Characteristics of patients with endometriosis and control group.

Patients		Control
N	100	100
Mean age (range) years	36.4 (23–58)	38.3 (29–61)
Body mass index (BMI) (kg/m^2^)		
<25	44 (44%)	42 (42%)
25 ≤ BMI < 30	48 (48%)	49 (49%)
≥30	8 (8%)	9 (9%)
Parity > 0		
yes	46 (46%)	49 (49%)
1	29 (29%)	31 (31%)
2 and more	17 (17%)	18 (18%)
no	54 (54%)	61 (61%)
Spontaneous abortions		
yes (spontaneous)	10 (10%)	2 (2%)
no	90 (90%)	98 (98%)
Use of hormone replacement therapy (HRT)		
yes	21 (21%)	14 (14%)
no	79 (79%)	86 (86%)
Clinical stage (rASRM) *		
I	11 (11%)	
II	14 (14%)	
III	44 (44%)	
IV	31 (31%)	

* The Revised American Society for Reproductive Medicine.

**Table 2 ijms-23-11583-t002:** Correlation of expressions of lncRNA in patients and controls.

	Mann–Whitney U Test		
lncRNA	*p*	N-Patients	N-Controls
UCA1	0.845	100	100
MALAT1	0.170	100	100
TC0101441	0.850	100	100
H19	**0.003**	100	100

Bold results are important *p* < 0.05000.

**Table 3 ijms-23-11583-t003:** Correlation between lncRNA and the clinical staging of the patients with endometriosis (*n* = 100) by the rASRM classification.

	Spearman’s Correlation		
lncRNA	R-Spearman	The Two-Sample *t*-Test T (N-2)	*p*
UCA1 & staging	0.131	1.10	0.062
MALAT1 & staging	0.100	1.68	0.068
TC0101441 & staging	0.120	0.84	0.204
H19 & staging	**0.240**	**2.625**	**0.022**

Bold results are important *p* < 0.05000.

## Data Availability

All data and materials, as well as a software application, support the published claims and comply with field standards.

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
