# Peer review of "Analysis of Long Non-Coding RNA (lncRNA) UCA1, MALAT1, TC0101441, and H19 Expression in Endometriosis"

_ijms, 2022, doi:10.3390/ijms231911583_

Round 1

Reviewer 1 Report

1. The major limitation in the study is that no detailed methods for experimental validation have been carried out. Any of the experimental methods, including lncRNA immunoprecipitation, lncRNA pull-down, lncRNA northern blot analysis, lncRNA in situ hybridization, lncRNA knockdown should have been done.

2. Clinico-pathological data cannot be directed correlated with specific lncRNA expression levels.

3. Multiple differentially expressed lncRNA’s associated with endometriosis have been reported in literature. Why were only 4 lncRNA’s expression was chosen and focused for the present work needs to be explained.

4. Diverse statements have been reported about involvement of H19 in endometriosis. H19 has been mentioned as downregulated in the present manuscript. However, contrastingly, H19 has been claimed to be upregulated by Hudson et al (Hudson QJ, Proestling K, Perricos A, Kuessel L, Husslein H, Wenzl R, Yotova I. The Role of Long Non-Coding RNAs in Endometriosis. Int J Mol Sci. 2021 Oct 22;22(21):11425).

Author Response

Thank you for your review.

I would like to kindly ask you to reconsider the publication of our revised paper:

Analysis of Long Non-Coding RNA (lncRNA) UCA1, MALAT1, TC0101441 and H19 Expression in Endometriosis

I hereby provide responses to the reviewers and list the changes that have been made in the revised version of our paper.

Rev 1

The major limitation in the study is that no detailed methods for experimental validation have been carried out. Any of the experimental methods, including lncRNA immunoprecipitation, lncRNA pull-down, lncRNA northern blot analysis, lncRNA in situ hybridization, lncRNA knockdown should have been done.

Thank you for your review. However, we dare to point out that our research was unfortunately limited in budget. Our Institute has allocated funding strictly for RT-PCR research. Of course, they will continue as the results are encouraging. In future work, we will try to include other experimental methods for validation.

  1. Clinico-pathological data cannot be directed correlated with specific lncRNA expression levels.

Sorry but we do not understand this comment. There are studies correlating expression with clinico-pathological data. We have published such a work with miRNAs

Romanowicz H, Hogendorf P, Majos A, Durczyński A, Wojtasik D, Smolarz B. Analysis of miR-143, miR-1, miR-210 and let-7e Expression in Colorectal Cancer in Relation to Histopathological Features. Genes (Basel). 2022 May 13;13(5):875.

  1. Multiple differentially expressed lncRNA’s associated with endometriosis have been reported in literature. Why were only 4 lncRNA’s expression was chosen and focused for the present work needs to be explained.

We selected lncRNA for research on the basis of literature data and especially work

 Maier IM, Maier AC.miRNAs and lncRNAs: Potential Non-Invasive Biomarkers for Endometriosis. Biomedicines 2021, 9, 1662.

The authors demonstrated the importance of four lncRNAs UCA1, MALAT1, TC0101441 and H19 in the development and pathology of endometriosis. Moreover, these lncRNAs have not been studied in the Polish population.

  1. Diverse statements have been reported about involvement of H19 in endometriosis. H19 has been mentioned as downregulated in the present manuscript. However, contrastingly, H19 has been claimed to be upregulated by Hudson et al (Hudson QJ, Proestling K, Perricos A, Kuessel L, Husslein H, Wenzl R, Yotova I. The Role of Long Non-Coding RNAs in Endometriosis. Int J Mol Sci. 2021 Oct 22;22(21):11425).

Thank you for your review. Our results may differ from those obtained by other authors as they were subject to certain limitations. Our population was small and we excluded infertile patients. According to literature data, the increased expression is in infertile women. Therefore, we treat the results with caution and want to continue the research.

I hope you find our revised Manuscript satisfying so that it can meet the criteria of publication in your Journal.

Looking forward to hearing from you,

Yours sincerely,

Beata Smolarz

Reviewer 2 Report

Nice paper. It should be published

Author Response

Rev2

Thank you for your review.

Best Regards

Beata Smolarz

Reviewer 3 Report

The paper submitted for review deals with a very important and at the same time challenging topic, which influences on many aspects of women's health. The manuscript is an original article analyzing the expression of four lncRNA in Endometriosis. The manuscript comprehensively presents the scientific data about expression of different lncRNA in Endometriosis with the special attention to UCA1, MALAT1, TC0101441 and H19. I only have a few minor comments. I am convinced that after taking these comments into account, the article can be published in IJMS

Introduction

1.     In 2021 the International Working Group of AAGL, ESGE, ESHRE and WES published "An international terminology for endometriosis". It would be good if the authors took into account the current terminology of the disease and cite a more recent publication in the introduction. This terminology was created in order to standardize the nomenclature of the disease and improve the quality of research on it. Taking this aspect into account may help improve the citation rate of this publication in the future.

2.     Paragraphs 4 - 7 should be moved to the Discussion section (from „Long non-coding RNA urothelial…” untill „…Treg and Th17 lymphocyte system”)

Materials and methods

1.     There is no information about the years for which data was collected.

2.     Table 1 should be moved to the Results section. 

3.  The information from the second paragraph regarding the age of the women participating in the study should also be included in the description of the group characteristics in the Results section (there is a lack of comprehensive description of this data in Resluts section). 

4.     Information on the age of women should be included in table 1. The authors describe this data in text, but this is not presented in the table 1.

5.     The first table should be corrected from the visual side. In the second line for "Group size" there should be "N" in parenthesis. What does the "-" in line 5 ("Parity") mean? In the line „Clinical stage” use rASRM in parenthesis and explain the meaning of the term directly below the table.

Results

1.     Again, I suggest to reanalize the visualization of the data contained in the tables. It seems to me that the data from Tables 2 and 3 should be combined into one table, taking into account the statistical significance between the groups (p-value). Additionally, column 2 (N) can be deleted. The methodology provides sufficient information about that. Additionally, this information is also included in Table 1. Alternatively, It also seems reasonable to change Tables 2 and 3 to graphs as in Figure 1 as they contain the same data. 

2.   Please make sure that data presented in Boxplot (figure 1) are presented correctly. Usually, boxplots contain data about the mean, quartiles 1 and 3, and extreme values. Following this interpretation, in the presented data in figure 1 (expression of H19) the line in the box should correspond to the mean value. However, it does not correspond to the value in the table 2 and 3. RQ mean value for H19 in control group is 72.5 while in boxplot this value is almost 200. Additionally, the extreme values according to the table are 0.104 and 810.33, while in the boxplot the extreme values are 0 and 600 respectively. I also propose to show the data in the form of boxplot for all lncRNAs, not just for H19. 

3.     Delete the last sentence in Figure's 1 caption. It should not include interpretation of the graph. 

4.     Control the abbreviations. Once the abbreviation rASRM has been expanded, there is no need to expand it a second time later in the text.

5.     There is no table 4 in the text.

6.   Table 5 does not present the data contained in the text, but only the correlations between lncRNA expression and rASRM. At the same time, it should be clarified which stages of endometriosis are concerned. According to the data contained in Figure 2, the stages were combined into groups. Of course, this is understandable, but needs to be described in the text.

7.     As in Figure 1, make sure that the data in Figure 2 is presented correctly.

Discussion

1.     After connection the informations from Introduction with Discussion section It is important to control presented statements and publications to avoid repetitions with the use of different citations. For example authors in the 5th paragraph of Introduction present the MALAT1 significance with the use of citation [11] and the same information is presented in paragraph 8 with the use of citation [38].

2.     A paragraph on limitation of the study is needed. The arguments are as follows: 

-       the control group is a group of women with uterine fibroids. Therefore, it is not a pure control group, and the result of the study rather analyzes changes in lncRNA expression between the group of women with endometriosis and the group of women with uterine fibroids. At least it needs to be included in the limitation section. 

-       Women with endometriosis are often treated before surgery, for example with hormonal contraception. This may affect the results. Please determine if this potential bias was controlled.

-       In 5th paragraph of discussion section, the authors indicate that: "H19 expression has been shown to be reduced in the endometrial tissue of infertile women”. Are women with infertility excluded from the study group? We know that endometriosis is diagnosed even in half of women with infertility. So isn’t there classification bias? It may turn out that the results of the study do not indicate a relationship between H19 expression in endometriosis but in infertility. 

Conclusions

1. The last paragraph from discussion section can be moved to conclusions.

Author Response

Rev2

Thank you for your review.

I would like to kindly ask you to reconsider the publication of our revised paper:

Analysis of Long Non-Coding RNA (lncRNA) UCA1, MALAT1, TC0101441 and H19 Expression in Endometriosis

I hereby provide responses to the reviewers and list the changes that have been made in the revised version of our paper.

the paper submitted for review deals with a very important and at the same time challenging topic, which influences on many aspects of women's health. The manuscript is an original article analyzing the expression of four lncRNA in Endometriosis. The manuscript comprehensively presents the scientific data about expression of different lncRNA in Endometriosis with the special attention to UCA1, MALAT1, TC0101441 and H19. I only have a few minor comments. I am convinced that after taking these comments into account, the article can be published in IJMS

Introduction

  1. In 2021 the International Working Group of AAGL, ESGE, ESHRE and WESpublished "An international terminology for endometriosis". It would be good if the authors took into account the current terminology of the disease and cite a more recent publication in the introduction. This terminology was created in order to standardize the nomenclature of the disease and improve the quality of research on it. Taking this aspect into account may help improve the citation rate of this publication in the future.

Answer. The article was cited in the section Introduction

  1. Paragraphs 4 - 7 should be moved to the Discussion section (from „Long non-coding RNA urothelial…” untill „…Treg and Th17 lymphocyte system”)

Answer The article has been redrafted in line with the comment

Materials and methods

  1. There is no information about the years for which data was collected.

Answer Information has been included in the section materials and method

  1. Table 1 should be moved to the Results section. The information from the second paragraph regarding the age of the women participating in the study should also be included in the description of the group characteristics in the Results section (there is a lack of comprehensive description of this data in Resluts section). 

Answer Has been corrected

  1. Information on the age of women should be included in table 1. The authors describe this data in text, but this is not presented in the table 1.

Answer Has been corrected

  1. The first table should be corrected from the visual side. In the second line for "Group size" there should be "N" in parenthesis. What does the "-" in line 5 ("Parity") mean? In the line „Clinical stage” use rASRM in parenthesis and explain the meaning of the term directly below the table.

Answer The table has been redrafted

Results

  1. Again, I suggest to reanalize the visualization of the data contained in the tables. It seems to me that the data from Tables 2 and 3 should be combined into one table, taking into account the statistical significance between the groups (p-value). Additionally, column 2 (N) can be deleted. The methodology provides sufficient information about that. Additionally, this information is also included in Table 1. Alternatively, It also seems reasonable to change Tables 2 and 3 to graphs as in Figure 1 as they contain the same data. 

Answer Tables and figures have been checked and redrafted

  1. Please make sure that data presented in Boxplot (figure 1) are presented correctly. Usually, boxplots contain data about the mean, quartiles 1 and 3, and extreme values. Following this interpretation, in the presented data in figure 1 (expression of H19) the line in the box should correspond to the mean value. However, it does not correspond to the value in the table 2 and 3. RQ mean value for H19 in control group is 72.5 while in boxplot this value is almost 200. Additionally, the extreme values according to the table are 0.104 and 810.33, while in the boxplot the extreme values are 0 and 600 respectively.I also propose to show the data in the form of boxplot for all lncRNAs, not just for H19. 

Answer Tables and figures have been checked and redrafted

  1. Delete the last sentence in Figure's 1 caption. It should not include interpretation of the graph. 

Answer Has been deleted

  1. Control the abbreviations. Once the abbreviation rASRM has been expanded, there is no need to expand it a second time later in the text.

Answer Has been corrected

  1. There is no table 4 in the text.

Answer Has been corrected

  1. Table 5 does not present the data contained in the text, but only the correlations between lncRNA expression and rASRM. At the same time, it should be clarified which stages of endometriosis are concerned. According to the data contained in Figure 2, the stages were combined into groups. Of course, this is understandable, but needs to be described in the text.

Answer Tables and figures have been checked and redrafted

  1. As in Figure 1, make sure that the data in Figure 2 is presented correctly.

Answer Tables and figures have been checked and redrafted

Discussion

  1. After connection the informations from Introduction with Discussion section It is important to control presented statements and publications to avoid repetitions with the use of different citations. For example authors in the 5th paragraph of Introduction present the MALAT1 significance with the use of citation [11] and the same information is presented in paragraph 8 with the use of citation [38].

Answer The article has been redrafted in line with the comment

  1. A paragraph on limitation of the study is needed. The arguments are as follows: 

-       the control group is a group of women with uterine fibroids. Therefore, it is not a pure control group, and the result of the study rather analyzes changes in lncRNA expression between the group of women with endometriosis and the group of women with uterine fibroids. At least it needs to be included in the limitation section. 

-       Women with endometriosis are often treated before surgery, for example with hormonal contraception. This may affect the results. Please determine if this potential bias was controlled.

Answer A paragraph on limitation has been corrected

-       In 5th paragraph of discussion section, the authors indicate that: "H19 expression has been shown to be reduced in the endometrial tissue of infertile women”. Are women with infertility excluded from the study group? We know that endometriosis is diagnosed even in half of women with infertility. So isn’t there classification bias? It may turn out that the results of the study do not indicate a relationship between H19 expression in endometriosis but in infertility. 

Answer We excluded infertile patients.

Conclusions

  1. The last paragraph from discussion section can be moved to conclusions.

Answer Has been corrected

I hope you find our revised Manuscript satisfying so that it can meet the criteria of publication in your Journal.

Looking forward to hearing from you,

Yours sincerely,

Beata Smolarz

Round 2

Reviewer 1 Report

1. Primer sequences used in real-time polymerase chain reaction assays haven't been mentioned.

2. Validation of the RT-PCR results using another platform is a must for discovery and validation of potential biomarkers. Else two different real time PCR platforms could have been used. Results stands inconclusive. 

3. Authors could have at least carried meta analyses of transcriptomic data freely available on NCBI's GEO for better insight and supporting their claim of H19 as a biomarker for endometriosis.

Author Response

Thank you for your review.

I would like to kindly ask you to reconsider the publication of our revised paper:

Analysis of Long Non-Coding RNA (lncRNA) UCA1, MALAT1, TC0101441 and H19 Expression in Endometriosis

I hereby provide responses to the reviewers and list the changes that have been made in the revised version of our paper.

  1. Primer sequences used in real-time polymerase chain reaction assays haven't been mentioned.

The primer sequences are presented in the materials and methods section

  1. Validation of the RT-PCR results using another platform is a must for discovery and validation of potential biomarkers. Else two different real time PCR platforms could have been used. Results stands inconclusive. 

In our opinion, RT-PCR is a very precise method. This technique is used in many research papers. The results of these works are accepted and published in recognized journals. Below we give examples of such works:

Wang W, Wu J. Identification of long noncoding RNA TC0101441 as a novel biomarker for diagnosis and prognosis of gastric cancer. Int J Clin Exp Pathol. 2021 Mar 1;14(3):363-368.

Kamrani S, Amirchaghmaghi E, Ghaffari F, Shahhoseini M, Ghaedi K. Altered gene expression of VEGF, IGFs and H19 lncRNA and epigenetic profile of H19-DMR region in endometrial tissues of women with endometriosis. Reprod Health. 2022 Apr 22;19(1):100.

 Li, Y.; Liu, Y.-D.; Chen, S.-L.; Chen, X.; De-Sheng, Y.; Zhou, X.Y.; Zhe, J.; Zhang, J. Down-regulation of long non-coding RNA MALAT1 inhibits granulosa cell proliferation in endometriosis by up-regulating P21 via activation of the ERK/MAPK pathway. Mol. Hum. Reprod. 2019, 25, 17–29

Huang, H.; Zhu, Z.; Song, Y. Downregulation of lncrna uca1 as a diagnostic and prognostic biomarker for ovarian endometriosis. Rev. Assoc. Med. Bras. 2019, 65, 336–341

Qiu, J.-J.; Lin, Y.-Y.; Tang, X.-Y.; Ding, Y.; Yi, X.-F.; Hua, K.-Q. Extracellular vesicle-mediated transfer of the lncRNA-TC0101441 promotes endometriosis migration/invasion. Exp. Cell Res. 2020, 388, 111815

We modeled ourselves on such works. In our research, RT-PCR was carried out in accordance with accepted standards. The expression of GAPDH served as an endogenous control. Each assay was performed in triplicate.

We believe that if RT-PCR is carried out in accordance with the required standards, it is a reliable test.

We have worked with this technique on other research papers and the results have been published flawlessly as to the methodology.

Bieńkiewicz J, Romanowicz H, Szymańska B, Domańska-Senderowska D, Wilczyński M, Stepowicz A, Malinowski A, Smolarz B. Analysis of lncRNA sequences: FAM3D-AS1, LINC01230, LINC01315 and LINC01468 in endometrial cancer.

BMC Cancer. 2022 Mar 29;22(1):343.

Zadrożna-Nowak A, Romanowicz H, Zadrożny M, Bryś M, Forma E, Smolarz B. Analysis of Long Non-Coding RNA (lncRNA) uc.38 and uc.63 Expression in Breast Carcinoma Patients.

Genes (Basel). 2022 Mar 28;13(4):608.

We have already written that in this work we focused only on this method for financial reasons. We treat our results very carefully, because we are aware of the limitations of our study.

  1. Authors could have at least carried meta analyses of transcriptomic data freely available on NCBI's GEO for better insight and supporting their claim of H19 as a biomarker for endometriosis.

In the discussion section, we discussed the importance of H19 in endometriosis (red font)

I hope you find our revised Manuscript satisfying so that it can meet the criteria of publication in your Journal.

Looking forward to hearing from you,

Yours sincerely,

Beata Smolarz

Round 3

Reviewer 1 Report

None